# A Long Way To Go:
# Investigating Length Correlations in RLHF

## Abstract

Great successes have been reported using Reinforcement Learning from Human Feedback (RLHF) to align large language models. Open-source preference datasets have enabled wider experimentation, particularly for "helpfulness" in tasks like dialogue and web question answering. RLHF in these settings is consistently reported to improve response quality, but also drives models to produce longer outputs. This paper demonstrates on three diverse settings that optimizing for response length is, more than previously suggested, a significant factor behind RLHF's reported improvements. We study the RL optimization strategies used to maximize reward, finding that improvements in reward are largely driven by length, often at the cost of other features. We find that even a *purely* length-based reward reproduces much of the downstream RLHF improvements over initial supervised fine-tuned models. Testing a diverse set of length-countering interventions across RLHF, we identify the dominant source of these biases to be the reward models, which we find to be influenced by strong length correlations in preference data that are learned during reward model training.

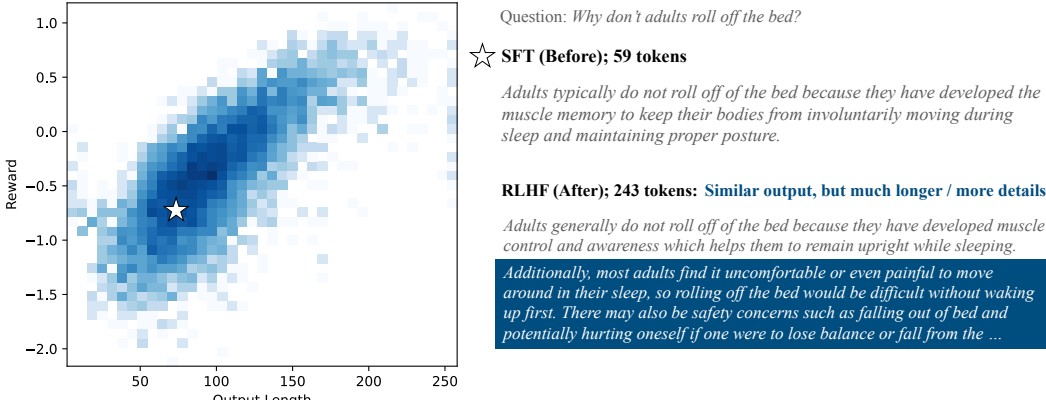

Figure 1: Log-scaled heatmap of output length vs. RLHF reward model score for a set of outputs generated from an SFT LLaMA-7B model on WebGPT. Reward correlates strongly with length, and running PPO consistently leads to longer outputs (right); this paper analyzes these phenomena.

## 1 Introduction

Reinforcement Learning from Human Feedback (RLHF) is widely used to align large language models (LLMs) with desired downstream properties such as helpfulness or harmlessness (Ouyang et al., 2022; Bai et al., 2022). Specifically, starting from supervised fine-tuned (SFT) LLMs, the standard paradigm generally consists of (1) training a *reward model* (RM) on a labeled set of preferences on pairs of outputs for the same inputs, and (2) optimizing the policy model with respect to the reward model via an RL algorithm like proximal policy optimization (PPO) (Schulman et al., 2017).

This procedure relies on two things. First, the reward model must be correctly specified and not misaligned with human preferences (Zhuang & Hadfield-Menell, 2021; Pang et al., 2022; Bobu et al., 2023). Second, the optimization algorithm must do a good job of balancing reward optimization with staying close to the initial policy distribution. Not meeting these conditions generally leads to

over-optimization of the reward model at the expense of human judgments (Dubois et al., 2023), which in the worst case leads to pathological "reward hacking" (Skalse et al., 2022). Ad hoc adjustments (Touvron et al., 2023b) and improvements in PPO (Zheng et al., 2023b) have stabilized the process, but there's still little work examining the changes in the policy model's behavior responsible for reward improvements, and to what extent these correspond to meaningful improvements in quality versus optimization of shallow reward correlations (Pang et al., 2022).

While the vast majority of "helpfulness" RLHF work has noted increases in *output length*, (Dubois et al., 2023; Zheng et al., 2023b; Sun et al., 2023; Wu et al., 2023; Nakano et al., 2021; Stiennon et al., 2020), little further analysis has been done, leaving the open possibility that length constitutes such a shallow correlation. Thus, seeking to better understand the objectives and improvements of RLHF, we focus on examining reward modeling and RLHF through the lens of length, finding it to play a larger role than previously discussed. We organize our findings into three parts:

(1) In two settings, nearly *all* PPO reward improvement comes from purely optimizing length. This holds even in the presence of diverse length-countering PPO interventions, suggesting reward modeling to be the source of length biases. (2) Studying training dynamics, we find these biases to originate from a combination of data imbalances and significant robustness issues in standard reward modeling. (3) We conduct an experiment where we measure how much doing PPO with a reward based *only* on length can reproduce PPO quality gains with trained reward models.

Optimizing length isn't necessarily bad: humans may genuinely prefer longer outputs. However, length's dominance comes at the cost of many other strong potential features. Further improvements to RLHF will likely require learning these other features, which itself requires strong solutions to the robustness flaws of reward modeling that we uncover: RLHF research still has a long way to go.

**Our Contributions:** (1) We propose several new analysis approaches and intervention strategies for RLHF (2) We conduct a multi-faceted exploration of length correlations at all stages of RLHF, demonstrating its effect to be much greater than previously anticipated (3) We identify underlying causes for said phenomena, particularly from robustness flaws of preference data and reward models (4) We plan to release code and a diverse set of models to support future open work.

## 2   TASK SETUP

RLHF is technique for optimizing the performance of text generation systems (Sutskever et al., 2014; Bahdanau et al., 2015), in which we place a distribution over target output $\mathbf{y} = (y_1, \ldots, y_n)$ given input sequences of words $\mathbf{x}$ via a generation model $\pi_\theta$: $p(\mathbf{y} \mid \mathbf{x}; \pi_\theta) = \prod_{k=1}^{n} p(y_k \mid \mathbf{y}_{<k}, \mathbf{x}; \pi_\theta)$. Historically, these models were trained with both language modeling pre-training (learning to predict the next word given context) and supervised fine-tuning (SFT; learning to generate outputs to maximize the likelihood of references on some dataset, also referred to as behavioral cloning).

RLHF is a technique introduced to further improve upon this approach, and can be broken into three components. First, it requires a set of **preference judgments** over model outputs of the form $P = \{(x_1, y_1^+, y_1^-), \ldots, (x_n, y_n^+, y_n^-)\}$ with triples of prompts $x_i$, preferred continuations $y_i^+$, and dispreferred continuations $, y_i^-$.

Then, given some $P$, the task is to train a scalar **reward model** $R(q, x)$ such that for any given preference triple, $R(x_i, y_i^+) > R(x_i, y_i^-)$. We use the standard Bradley-Terry preference model (Bradley & Terry, 1952), where $P(y_1 \succ y_2 \mid x) = \frac{\exp(R(x, y_1))}{\exp(R(x, y_1)) + \exp(R(x, y_2))}$ and the reward model is trained to optimize the log likelihood of the observed preferences.

Finally, given $R$, we use **reinforcement learning**, specifically proximal policy optimization (Schulman et al., 2017, PPO) to optimize a supervised fine-tuned (SFT) model $\pi_\theta^{\text{SFT}}$ to get a model $\pi_\theta^{\text{RL}} = \text{PPO}(\pi_\theta^{\text{SFT}}, R)$ that, for a query distribution $X = (x_1, \ldots, x_m)$, maximizes the reward $R(x_i, \pi_\theta(x_i))$, with a constraint that we not deviate too strongly from the initial distribution. RL optimization in PPO is based on the maximization of the following equation, where $\lambda$ controls the strength of a Kullback-Leibler (KL) divergence penalty between the original policy $\pi_\theta^{\text{SFT}}$ and the current policy $\pi_\theta^*$ at a given step. :

$$R_{\text{final}}(x, y) = R(x, y) - \lambda D_{\text{KL}}(\pi_\theta^*(y|x) \| \pi_\theta^{\text{SFT}}(y|x)) \tag{1}$$

## 2.1 TASKS

We explore a collection of three preference datasets corresponding to three tasks and preference labeling strategies (examples in Appendix C). We select these datasets to provide a diversity of "helpfulness" tasks that are still challenging for our base model, LLaMA-7B (Touvron et al., 2023a).[1]

**WebGPT (Question answering; human labels)**   This dataset (Nakano et al., 2021) contains human annotated preference labels between two outputs for the open-domain long-form question answering (LFQA) task (Fan et al., 2019). As human annotation is expensive, this dataset is relatively smaller at only 19.6K examples (mean tokens per $y = 169$) compared to the others we study.

**Stack (Technical question answering; upvotes)**   Released by Hugging Face, this dataset collects technical questions and answers from StackExchange (Lambert et al., 2023). The preference label between two answers is derived using the number of upvotes; the one with more upvotes is assumed to be preferred. We use a subset of 100K (mean tokens per $y = 236$) pairs from the dataset following the Hugging Face implementation (von Werra et al., 2020).

**RLCD (Multi-turn conversation; synthetic preferences)**   Finally, we explore multi-turn dialogue style data, released by Yang et al. (2023). Starting from the input instructions in the Helpful/Harmless dataset by Anthropic (Bai et al., 2022), they automatically generated preferred and not-preferred outputs using prompt heuristics, e.g. appending "generate unhelpful outputs" to the prompt. The "helpfulness" subset that we use consists of 40K examples and mean tokens per $y = 45$.

## 2.2 EXPERIMENTAL SETUP

**Framework**   We use the standard implementation and hyperparameters for the 3 components of RLHF to maintain consistency. We base our RLHF implementation on the Huggingface TRL framework with hyperparameters we find to work best based on reward convergence and downstream evaluation ($\lambda = 0.04$, batch size 64, see more details in Appendix A) (von Werra et al., 2020), and use LoRA (rank=16) (Hu et al., 2021) to enable training large Llama-7B models (Touvron et al., 2023a) with limited GPU memory. For our SFT models we use the released AlpacaFarm SFT model for WebGPT and RLCD as we find it to work well, and the TRL SFT model for Stack.

**Evaluation**   Our evaluation relies on two factors. First, **reward** is an intrinsic metric optimized by the PPO process. Second, we follow past work in AlpacaFarm (Dubois et al., 2023) to conduct downstream evaluation using more powerful LLMs as proxies for human preferences. Specifically, we sample responses on fixed held-out test sets of 500 prompts for each setting, then use their exact evaluation scheme based on using a panel of 12 simulated OpenAI API based "annotators," which they show correspond well with human preference judgements. The final format is an overall pairwise "win rate" of one set of paired outputs vs another, which we call **simulated preferences**.

## 3 EXAMINING PPO

In this section, we first show that: (1) Output length increases during PPO (Figure 2). (2) There exists a positive correlation between length and reward model scores (Figure 3). Taken together, this evidence suggests that simply increasing length *could* be a successful way to improve reward. Motivated by this, we investigate the following question: Is length increase the *primary* factor for reward models scores increasing during PPO, or are other features also optimized?

## 3.1 LENGTH INCREASES DURING PPO

To contextualize the rest of the work, we first show that length actually *does* increase as a result of PPO. Indeed, when comparing histograms of generation lengths (see Figure 2) on a fixed query set before and after our initial PPO runs, we find that PPO causes notable length increases.

---

[1]Note: Our settings are oriented towards helpfulness, which we infer to be closer related to length, however studying our approaches on other objectives such as harmlessness could be interesting future work.

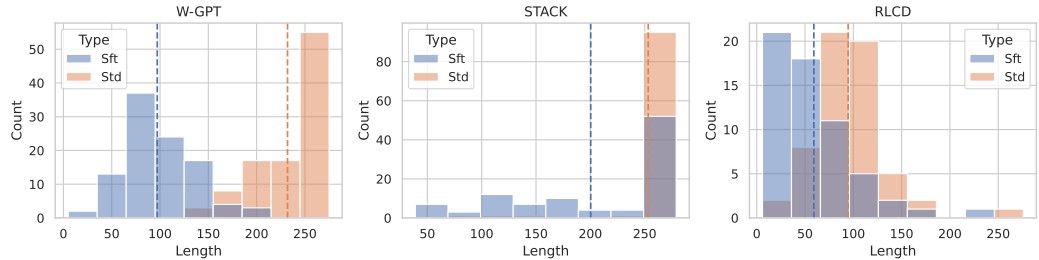

Figure 2: Histograms of output lengths of SFT model before (blue) and after (red) standard PPO (STD); averages shown with dashed lines (a) Across settings, PPO leads to dramatic length increases

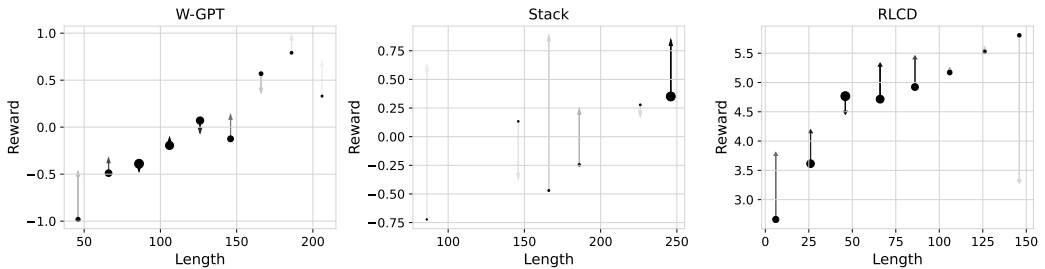

Figure 3: Scatter plots with output length vs reward in 20 token buckets. Arrows indicate improvement (up) or degradation (down) of reward by doing PPO (HIGH KL) within a bin. Size and color intensity scaled by proportion of examples in bin (a) Reward has visible correlations with length (b) On WebGPT and RLCD, reward improvement within bins is much smaller than reward improvement from shifting to the right: PPO's barely optimizing non-length features. Quantified in Table 1

We now investigate the extent to which *other* features are learned, with two different settings of the KL weight $\lambda$ in the objective. Figure 3 shows reward scores stratified by length, binned into buckets of 20 tokens for the higher $\lambda$ variant (HIGH KL). While reward score does increase in each bin on average, the increases in reward are uneven. Furthermore, the increases are less strong than the length trends: generating an answer that's 40 tokens longer (shifted over by two bins) often provides a larger improvement than PPO. (See Figure 10 for a plot with our standard, lower-KL PPO setting.)

To quantify this more precisely, we estimate the percentage of length-based optimization as the *ratio* of *weighted reward gain* (WRG) to the overall *reward improvement* ($\Delta R$) from PPO, where weighted reward gain is the sum of each bin's difference value multiplied by the total number of examples in each bin. Weights are computed by total examples from SFT and PPO combined.

Table 1: Weighted reward gain (WRG), reward impovement ($\Delta R$), and their ratio, for PPO with standard (STD) and high (HIGH KL) $\lambda$ (a) Low ratios on WGPT and RLCD indicate high PPO dependence on length. STACK shows this pattern to a weaker extent (b) similarity of STD, HIGH KL indicates invariance of this dependence to interventions (Section 3.2)

|  | WGPT | | STACK | | RLCD | |
|---|---|---|---|---|---|---|
|  | STD | HIGH KL | STD | HIGH KL | STD | HIGH KL |
| $\Delta R$ | 0.82 | 0.20 | 0.89 | 0.67 | 0.94 | 0.61 |
| WRG | 0.02 | 0.03 | 0.48 | 0.37 | 0.25 | 0.12 |
| ratio | 2.0% | 15.1% | 53.4% | 56.5% | 27.2% | 19.1% |

Table 1 reports results. Revisiting this in the context of Figure 3, we see that around **70%–90% of the improvement on WebGPT and RLCD is explained purely by shifts in length**. STACK shows a lower value here, with only about 40% of the gain arising from length. One reason for this is that STACK outputs are close to the length limit during training,[2]

---

[2]Stack, due to SFT having higher initial length, tends to generate unboundedly long outputs after PPO. We set a higher max length (216) than the source TRL codebase (128) for Stack; however the pattern remains.

Table 2: Tokens (LEN), reward RM, simulated preference (SIM PREF, Section 2.2) vs. standard PPO (STD) across interventions (blue if better, red if worse than STD). Columns with failed reward optimization excluded. [*] for statistically significant delta from STD with $p < 0.05$, bootstrap test (a) Interventions only mitigate length increases vs SFT starting point, that too cost to reward: Length-based optimization invariant to RL interventions, reward modeling is likely more important

| | W-GPT | | | | | STACK | | | | RLCD | | | | |
| | SFT | STD | RM-SC | H-KL | OMIT | SFT | STD | RM-SC | H-KL | SFT | STD | RM-SC | LEN-C | H-KL |
|---|---|---|---|---|---|---|---|---|---|---|---|---|---|---|
| LEN | 100 | 230 | 128 | **120** | 127 | 203 | 257 | 249 | 250 | 59 | 94 | 82 | **72** | 97 |
| RM | -0.45 | **0.25** | -0.05 | -0.06 | -0.13 | 0.05 | **0.74** | 0.40 | 0.30 | 4.4 | **5.50** | 5.00 | 5.20 | 5.20 |
| SIM PREF | 42%[*] | – | 49% | 45%[*] | 48% | 42%[*] | – | 46%[*] | 45%[*] | 37%[*] | – | 41%[*] | 44%[*] | 43%[*] |

so gain from increasing length is not possible to achieve. Second, Stack's technical QA setting represents a different style of answer that we believe *does* require optimizing for features beyond length.

## 3.2 INTERVENING ON OPTIMIZATION

We see that in a *standard* pipeline, PPO has a tendency to optimize only on length, but what if we constrain optimization to mitigate this? We test the effects of several interventions below.

The simplest intervention to PPO to encourage short outputs is to just **increase the KL coefficient $\lambda$ (H-KL)** (Equation 1), with the intuition that closer to the initial distribution should mean closer to the initial length. We experiment with setting it to 0.12 instead of 0.04; larger values impede model convergence.

We also experiment with a scalar penalty on the reward to **control length (LEN-C)**. We set $R' = \sigma \left(1 - \frac{\text{len}(y)}{N}\right)$, where $N$ is a maximum length value that we do not want PPO to exceed, and $\sigma$ is a moving average of batch reward standard deviation.[3]

A similar option to prevent outputs from getting longer may just be to altogether **omit (OMIT) outputs beyond a length threshold** from PPO, so that no update is made to encourage these. In practice we swap these examples with randomly sampled outputs from the batch.

Finally, prior work examining ways to improve implementations of PPO mentions that **reward scaling (RM-SC)** can be useful for "controlling training fluctuations" and reducing over-optimization (Zheng et al., 2023b). Similar to batch normalization (Ioffe & Szegedy, 2015), for each batch $X, Y$ of sampled outputs, we compute the mean ($\mu$) and standard deviation ($\sigma$) of $R$. We then take a moving average of these values across $N$ previous batches and "scale" $R$ to become $R' = \frac{R-\mu}{\sigma}$, where we note $\sigma$ remains relatively constant across training.

**Results** We report results for the interventions on the reward score and PPO in Table 2. Note the RM row is comparable within each setting since we use the same underlying reward models, and thus we use it as our primary metric to reason about length and reward tradeoffs. We also report simulated preferences (see Section 2.2) vs STD, where $< 50\%$ indicates being worse than standard PPO on downstream answer quality.

We find that across all interventions, **length always increases relative to SFT**, and **reward model score is always worse** than standard PPO. These patterns suggest that a strong component of PPO *is* related to length. Including the fact that length control (LEN-C) led to convergence failure (reward not increasing during training) on W-GPT and STACK, this suggests that length is a difficult feature to disentangle post-hoc from reward.

Recalling the scatter plots from Figure 3, we note that across all of these different interventions, the scatter plots display similar patterns (see Appendix B), implying that while these interventions reduce the overall optimization towards length, they *don't* change the fundamental tendency of PPO

---

[3]We try several variants of this idea, such as a scalar penalty past a length threshold, and note similar convergence failures. In general, we find that stricter versions of these constraints negatively affects convergence.

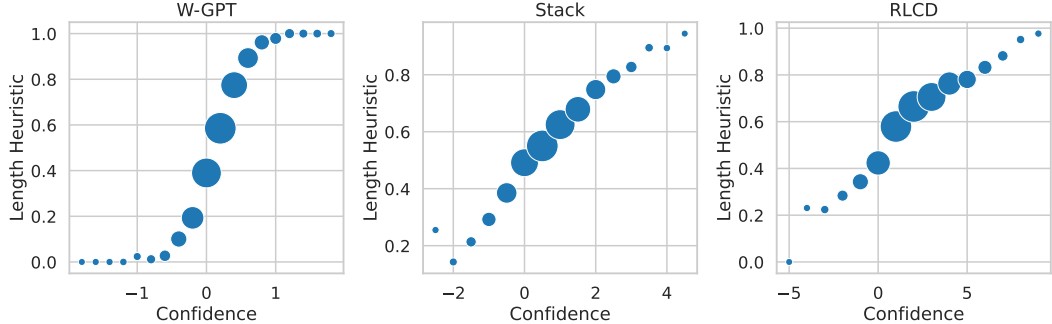

Figure 4: Training example confidence ($c_i$) vs length heuristic, bucketed based on $c_i$, size shows amount in bucket (a) Most examples are in the low-confidence center: RMs have trouble learning on most data (b) Really strong predictions (including incorrect) follow length heuristic with clean proportionality: RMs learn strong features from a small set of "easy" length-following examples

to avoid optimizing for other features. However, while length still increases with respect to SFT, several interventions *do* allow for length increases to be mitigated while still recovering a large portion of reward and downstream performance gain (e.g., RM-SC).

## 4 EXAMINING REWARD MODELING

Section 3.2 showed that interventions during PPO do not fully mitigate the issue of reward gains coming from length increases. We now investigate whether we can intervene even earlier in the process, on the preference data itself, in order to circumvent this length dependence.

### 4.1 ANALYZING PREFERENCES OVER TRAINING

One root cause of length correlation is length imbalances in the preference datasets, where longer answers are systematically preferred to shorter answers. We can measure this with **length heuristic agreement**: the accuracy of always predicting that the longer output is the gold preferred output (see Table 3): We see that all datasets are slightly imbalanced towards longer outputs. However, this doesn't fully explain the strong correlations suggested earlier in Figure 3.

Table 3: Preference eval accuracy of always preferring longer response. Above random (50%) accuracy indicates data length bias

| WGPT | STACK | RLCD |
|------|-------|------|
| 55.7% | 59.6% | 63.1% |

To understand this better, we can study *training dynamics* of reward model learning by computing statistics over several epochs of training. Given reward model $R$ being trained on preference dataset $P$ for $E$ epochs, we can track each data point $(x_i, y_i^+, y_i^-) \in P$ where we compute the distribution of *confidence* (RM score of "preferred" subtracted from "dispreferred"), at each epoch $c_i = \{(e, R(x_i, y_i^+) - R(x_i, y_i^-)) : e \in \{2, \ldots, E\}\}$, where we exclude epoch 1 to mitigate noise.

**Results** First, we note that when examining "cartography" plots (Swayamdipta et al., 2020) examining the mean ($\overline{c_i}$) and variance ($\sigma(c_i)$) of different $c_i$ (see Appendix B.2), we find that the values are largely centered at zero, suggesting that reward models are not able to make progress on most training examples: the predictions are low-confidence and largely do not change. This suggests that most features are instead learned on the set of "easy" examples with higher $\overline{c_i}$.

With the hypothesis that length may be related to "easy" examples, we use length heuristic accuracy again, but this time, we compute it on slices where we bin training examples based on $\overline{c_i}$, plotting these bins by confidence (x-axis) against length heuristic accuracy (y-axis) on each slice as scatter plots in Figure 4.

The figure shows strikingly clean patterns, with the mean confidence $\overline{c_i}$ for data in an interval of training examples correlating strongly with the length heuristic. This means that (1) the length heuristic applies to most examples that are easy, and (2) perhaps more tellingly, **the overwhelming majority of "hard" examples are cases where the model follows the length heuristic to confi-**

**dently predict the wrong answer**. Overall, this supports that length is one of the strongest features learned in these models. Note that WebGPT, with the strongest pattern, also displayed the lowest WRG from Table 1, implying that these correlations propagate through all stages.

### 4.2 INTERVENTIONS ON PREFERENCE DATA

#### 4.2.1 SETUP

Given the strong length biases learned from preference data in standard RMs (STD), we now examine whether we can eliminate these biases by strategically modifying preference data.

**Length Balancing (BAL)**    The simplest intervention is to remove length biases from the preference data. Specifically we balance data such that the distribution of pair length differences are symmetric by bins of 10. Suppose there are more examples where preferred responses are 20 tokens longer than dispreferred ones compared to the reverse case; we then subsample the cases which are 20 tokens longer until they match the number of cases which are 20 tokens shorter, thereby balancing the data.

**Confidence-Based Truncation (C-TR)**    Our previous results suggest that something more data-specific beyond a surface length bias may influence training: for example, a particular set of "easy" examples may be corrupting the data, and removing them may help, as established in literature on dataset cartography Swayamdipta et al. (2020). Given that we've trained some $R_{\text{base}}$, and computed $\overline{c_i}$ on dataset $P$ (Section 4.1), we can test this idea by training a new RM $R_{\text{trunc}}$ on a subset of $P$ where $\overline{c_i} < \theta_1$ and $\overline{c_i} > \theta_2$, with threshold hyper-parameters $\theta_1$, and $\theta_2$. We experiment with several variants (see Appendix B.2), keeping sets of 50% of the data for each. Below we report results when we set $\theta_1 < \theta_2$, keeping a central subset of data.

**Reward Data Augmentation (R-DA)**    In line with the hypothesis that over-optimization stems from spurious correlations in the data, another potential intervention is data augmentation, specifically using "random pairing" where we can pair matching prompt output pairs $q_i, p_i^-$ from $P$ with $p_i^-$ serving as a "prefered" example, and a randomly sampled $p_j^+$ from another prompt serving as a "dispreferred" example. This serves to encourage disregarding stylistic features in favor of relevance to the query.

Table 4: Eval accuracy (ACC) and pearson within batch (CORR) for RM interventions (RAND is random baseline, STD normal RM). (a) Most approaches either reduce correlation *or* maintain good accuracy, but very few do *both*: length bias is highly tied to RM success

|  | WGPT | | STACK | | RLCD | |
|---|---|---|---|---|---|---|
|  | ACC | CORR | ACC | CORR | ACC | CORR |
| RAND | 50% | 0 | 50% | 0 | 50% | 0 |
| STD | 61.5% | 0.72 | 70% | 0.55 | 80% | 0.67 |
| BAL | 52.6% | **-0.13** | 61.9% | **-0.09** | 73.1% | 0.62 |
| C-TR | 58.8% | 0.67 | 59.5% | 0.31 | 77.2% | 0.57 |
| R-DA | **62.5%** | 0.35 | **72.6%** | 0.37 | **80%** | **0.43** |

#### 4.2.2 RESULTS

We first report in Table 4 the evaluation accuracy of these different reward models, as well as a **correlation within batch** (CORR) measure which, given sets of 8 generations, is the mean Pearson correlation between output length and reward model score for each batch. While the standard reward model (STD) achieves high accuracies across settings, this comes with high length correlation.

Data Augmentation (R-DA) improves on both of these partially, while confidence-based truncation (C-TR) brings length correlation down at the cost of accuracy. Note that, when using correlation within batch, we find that BAL leads to length bias being reversed, but at near-random accuracies, while other truncation strategies don't yield notable differences. These patterns indicate that, perhaps because RMs fail to learn on most examples, they are particularly brittle, and can learn spurious correlations easily. As the only setting where length balancing eliminates correlation and maintains above-random accuracy, we see more evidence that STACK is the one setting of our three where reward models can learn features independent from length.

We then show results for downstream adjustments to preference data in Table 5: Length still usually increases from the SFT starting point, though many interventions are shorter relative to STD. BAL

Table 5: Simulated preference (SIM PREF) vs STD PPO for the SFT model, length (LEN), STD PPO, and interventions (a) STACK BAL shows strong results possible without length increase via RM interventions (more influential vs PPO interventions), but RMs seem brittle, and results are inconsistent

| Method | W-GPT | | | | STACK | | | | | RLCD | | | | |
|---|---|---|---|---|---|---|---|---|---|---|---|---|---|---|
| | SFT | STD | R-DA | C-TR | SFT | STD | BAL | R-DA | C-TR | SFT | STD | BAL | R-DA | C-TR |
| LEN | 100 | 230 | 139 | 141 | 203 | 257 | 148 | 256 | 244 | 59 | 94 | 82 | 112 | 97 |
| SIM PREF | 42%[*] | – | 49% | 44%[*] | 42%[*] | – | 57%[*] | 58%[*] | 44%[*] | 37%[*] | – | 44%[*] | 44%[*] | 50% |

Table 6: Simulated preferences (against SFT and STD PPO) from *purely* optimizing for higher length (LPPO), with and without ($\lambda = 0$) KL penalty, and a longest-of-8 sampling baseline (SFT-LONG) (a) LPPO is comparable to STD PPO: further supports hypothesis that RLHF is largely length-based (b) interestingly LPPO beats $\lambda = 0$, SFT-LONG even when shorter: KL-constrained PPO with just length causes qualitative changes beyond just extending output length

| | W-GPT | | | STACK | | | RLCD | | |
|---|---|---|---|---|---|---|---|---|---|
| | SFT-LONG | LPPO | LPPO $\lambda = 0$ | SFT-LONG | LPPO | LPPO $\lambda = 0$ | SFT-LONG | LPPO | LPPO $\lambda = 0$ |
| LEN(SFT) | 100 | – | – | 203 | – | – | 59 | – | – |
| LEN | 141 | **118** | 167 | **249** | 252 | 248 | 117 | **98** | 163 |
| SIM PREF (PPO) | – | **48%** | 47% | – | **43%**[*] | 42%[*] | – | **48%** | 44%[*] |
| SIM PREF (SFT) | 48% | **56%**[*] | 53% | 57%[*] | **59%**[*] | 58%[*] | 52% | **64%**[*] | 51% |

on STACK, perhaps due to there being other easy non-length features to learn, even leads to shorter outputs than SFT, confirming the importance of preference data to final PPO length biases.

Unlike our PPO interventions described in Table 2, simulated preference doesn't always decrease with preference data interventions: On STACK, where BAL is shorter than SFT, it *also* improves SIM PREF over normal PPO, suggesting that at least in noisier settings there is somehow room for PPO to do more than just increase length, but this pattern is inconsistent. Compared to later stages, interventions on preference data seem to be the most promising for overall improvement of RLHF beyond length, though the fundamental inability of reward models to learn well from data remains.

## 5  HOW FAR CAN LENGTH GO?

Many of our experiments suggest that our reward models are primarily guiding PPO to produce longer outputs, yet we still see improvements on downstream simulated preferences. One explanations for this is that humans and models like GPT-4 have a bias towards preferring longer outputs in the settings we study (Zheng et al., 2023a). Another possibility is that optimizing for length with PPO intrinsically improves the quality of generation even in the absence of other features.

We investigate two interventions aimed *purely* at increasing length, which show how far optimizing for this single aspect can go. First, we sample 8 outputs from the SFT model and choose the longest one (**SFT-LONG**). Second, we use **length as our reward** for PPO (keeping the standard KL term) with $R^*(y) = 1 - \left| \frac{len(y)}{N} - 1 \right|$. In this case, $N$ is a target length hyperparameter (set to 156, 120, and 200 on WebGPT, RLCD, and STACK respectively). We call this setting LPPO, and also explore a variant of length-only PPO with $\lambda$ set to 0 (LPPO $\lambda = 0$) in Table 6.

First, we note that SFT-LONG can lead to moderate improvements (57% winrate vs SFT on STACK and 52% on RLCD), though not on WebGPT. When we then compare to LPPO, we find that **purely optimizing for length actually reproduces most of the performance improvements of RLHF** with the reward models. Notably, this approach yields simulated preference improvements over SFT-LONG, which has even longer outputs.

It is still possible that RLHF with our reward models *does* lead to other changes or improvements in the outputs beyond length. This experiment also does not necessarily establish flaws in the preference judgments; these outputs with the right length are often more informative and more useful (Figure 1). However, it shows that downstream gains *can* be explained by optimizing for length.

## 6    RELATED WORK

**RL**    Reinforcement learning from human feedback has been explored extensively (Knox & Stone, 2009), often being used in robotics tasks to extrapolate reward signal beyond an initial preference set (Brown et al., 2019). Recent work in NLP has explored implementations (Zheng et al., 2023b; Touvron et al., 2023b), objectives (Wu et al., 2023), and even alternatives (Rafailov et al., 2023; Zhao et al., 2022; 2023) for RLHF, but have generally overlooked or dismissed length increases. Our work is largely orthogonal to these directions, using the issue of length to analyze the lack of robustness in current reward models. Finally, other past uses of RL in NLP (Ammanabrolu & Riedl, 2018; Martin et al., 2017; Ramamurthy et al., 2023) have largely faced different sets of issues due to reward not coming from models learned over human preferences.

**Reward Model**    In the context of noisy and biased preference data, are reward models able to learn robust features reflecting the underlying preferences? In broader NLP, dataset artifacts have been a prevalent issue even on simpler settings like natural language inference (Gururangan et al., 2018; Poliak et al., 2018). In the context of RLHF, Stiennon et al. (2020) notes that over-optimizing for a reward model leads to pathological summaries, Dubois et al. (2023) notes a pattern of human preferences going up briefly then down as reward model score increases, and Pang et al. (2022) present some cases where reward hacking can be produced within synthetic settings. Our work, in comparison, delves further into what causes reward over-optimization in *realistic* settings, while also further exploring diagnostics and solutions. We focus on length as it is the most prevalent, but our experimental paradigm is applicable to any analysis of over-optimization in RLHF.

**Length control and length biases**    Techniques outside of RLHF for controlling length of NLP models have been explored (Kikuchi et al., 2016; Ficler & Goldberg, 2017). Length divergences specifically between training time and test time have been explored in the machine translation literature (Riley & Chiang, 2022), but these have been attributed to inference techniques and label bias in text generation methods. The open-ended nature of our generation problems is quite different from MT. Murray & Chiang (2018) use a per-word reward similar to our per-word penalty in RL, though to solve the opposite problem of outputs being too short. Finally, in discriminative "text matching" tasks like paraphrasing, past work has observed similar length heuristics, Jiang et al. (2022), but the sentence-pair format of these tasks makes their issues somewhat different.

## 7    CONCLUSION AND LIMITATIONS

In this work we study correlations of length and reward in RLHF. Across three datasets and several stages of observational and intervention-based exploration, we make a case that RLHF in these settings achieves a large part of its gains by optimizing for response length.

While the extent of the patterns we find are surprising, this doesn't necessarily invalidate the potential of RLHF. We note that our Stack setting, which involves the most technical responses, does demonstrate improvements in reward even for outputs already at our maximum length. Furthermore, optimizing purely for length *does* seem to lead to "qualitative" improvements beyond just sampling from the base model and choosing longer outputs, indicating that the learning dynamics of RLHF may be beneficial for LM training. Rather than claiming length to be an inherent shortcoming, we seek to use it as a vehicle to analyzing RLHF's successes and failures.

One limitation of our work is that, while we explore diverse settings, we are restricted to open-source preference datasets. Recent work such as Llama-2 (Touvron et al., 2023b) develops an extensive dataset of preferences and pursues a sophisticated RLHF strategy, which may not face the limitations we do. Furthermore, we focus primarily on a broad "helpfulness" objective (again, aligning with these preference datasets) using LLaMA-7B as the base model. While these represent a substantial fraction of research on open reward models, our findings may not necessarily apply to RLHF running on larger closed-source models, or with alternate objectives like "harmlessness".

Despite these limitations, we believe our work shows that RLHF with these reward models is not yet achieving its full potential. We believe that developing more accurate and robust reward models, either by changing the reward model, its objective, or the preference collection process, may hold the key to unlocking the full capabilities of RLHF.

## REPRODUCIBILITY

For our various studies on the relationship between RLHF and length, we first trained a set of reward models and policy models. In order to support future open RLHF research, we release our code as well as reward and policy models. In addition to detailing our experimental setup and evaluation scheme in Section 2.2, as well as describing our interventions in detail in Section 3.2 and Section 4, we include further hyper-parameters and instructions in Appendix A. Note that we use open preference datasets, publicly available base models, and open-source RLHF code that doesn't require prohibitive computational resources.

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

## A  TRAINING / EVALUATION DETAILS

**Hardware**  All experiments were conducted across 2 workstations, one with 4 NVIDIA RTX A6000 GPUs and another with 8 NVIDIA A40 GPUs. However, all of our individual experiments were run across 2 GPUs. In this configuration, training an RM takes around 6 hours on the Stack dataset, 3 hours on the RLCD dataset, and 1 hour on the WebGPT dataset after Ĩ epoch. For PPO, training takes around 12-18 hours for a single run.

### A.1  REWARD MODELS

For StackExchange, we have a train set of 100K examples and an evaluation set of 10K examples. For WebGPT, since we use 18k examples from training and 1.6K as an evaluation set. For RLCD, we use 40K training examples and 2.6 examples for the test set, where we use these test sets in all of the evaluation we show above.

For training, we follow a rule of continuing to train until eval accuracy stops going up. Prior work finds that reward model training for just 1 epoch is most effective to avoid over-fitting, however for

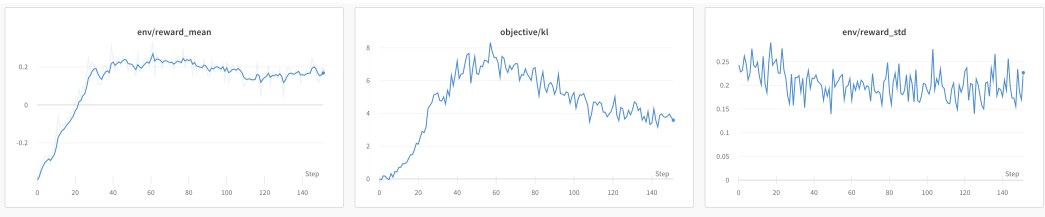

Figure 5: Training for standard WebGPT run

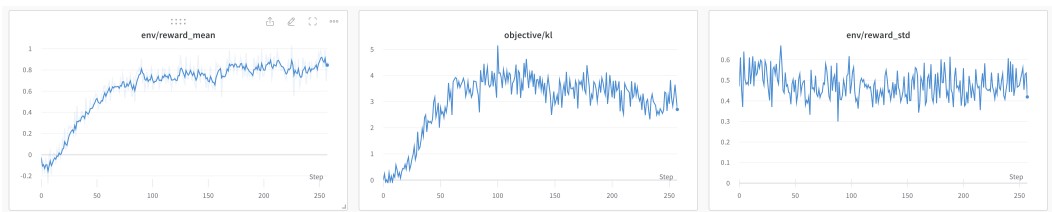

Figure 6: Training for standard Stack run

some of our preference data interventions we note that convergence takes longer. Overall, this ends up with usually 1-2 epochs of training at most for the checkpoints that we use. We use bfloat16, learning rate of 1e-5, and batch size of 2 with 2 gradient accumulation steps. With these configurations, 1 epoch on 10K examples takes around 2 GPU hours.

Note that for the training dynamics analysis (Figure 4), we run reward model training for 5 epochs to reduce variance, however we don't use those models directly (though we note that eval accuracy doesn't go down significantly even at that point).

## A.2 PPO

For our RLHF step, as stated before, we use LoRA and 8-bit quantization for the policy and reward models, since the TRL training configuration requires having all used models on each device used for training. We merge reward model and generation models with LoRA adapters before PPO.

Past work has commented on the stability of PPO and "secrets" needed to get it working well (Zheng et al., 2023b). We found that setting the right KL coefficient and batch size were the most important for stable convergence.

For training we generally run training for between 150-200 steps, where this is a hyperparameter for each dataset depending on speed of convergence and allowing sufficient steps for KL to decrease in certain settings (Figure 5). We experimented with runs of up to 400 steps and generally did not find improvement in simulated preference or reward.

With 2 GPUs, batch size of 32 on each, training takes around 16 hours to complete 200 steps, giving an overall time of 32 GPU hours per PPO model. Note that we use max length of 156 on WebGPT and RLCD, and 216 on STACK since it has a higher starting point for the SFT model (note that this hyperparameter has a strong influence on training speed).

Figures 5, 6, 7 show statistics over the course of training for our standard settings, with KL of 0.04. We note that RLHF does successfully increase reward score. The last half of training usually yields a decrease in KL divergence, as the model has optimized for reward and is regularized closer to the initial policy model by the KL term.

## A.3 INFERENCE / EVALUATION

Once we have our trained PPO models, we finally sample outputs that we can use to compare different systems and evaluation. For all results, unless otherwise stated, we generate 500 outputs each from a fixed set of the held out data from each dataset, and base our results on those (we find this to be a sufficient amount, especially when comparing patterns across a set of interventions /

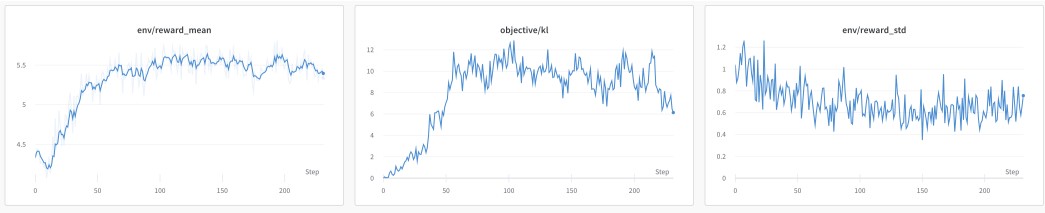

Figure 7: Training for standard RLCD run

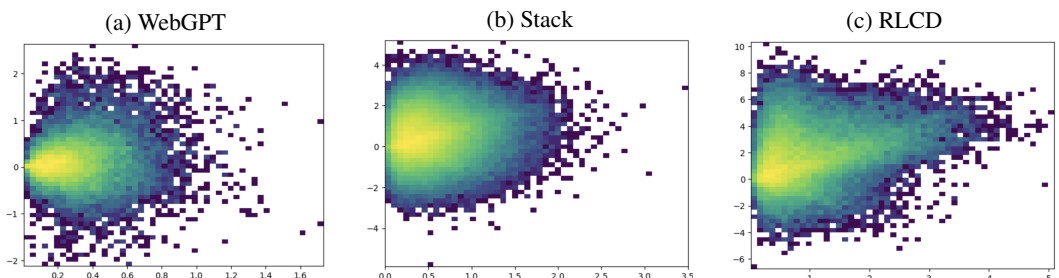

Figure 8: Logarithmically scaled heatmaps plotting, with a preference pair in training data being a single point, the variance of confidence over training (x-axis) vs the mean confidence (y-axis). These are the plots for WebGPT, STACK, and RLCD respectively left to right.

settings which themselves serve as additional datapoints). Computing simulated preference (Dubois et al., 2023) for 100 outputs costs around $3.5 USD using the OpenAI API. These calls are made to gpt-4-0314, gpt-3.5-turbo-0301, and text-davinci-003.

We decode with nucleus sampling (Holtzman et al., 2020) with a $p$ value of 0.9, maximum length of 256, temperature 0.9, and a repetition penalty of 1.2 (based on the TRL repository default hyperparameters). Whenever we sample multiple outputs from a single prompt, we draw 8 samples.

### A.4  Interventions

For length control, we set the length center $N$ to the starting mean SFT length, noting similar patterns with different configurations as well (50 tokens for RLCD, 100 for WGPT, 200 for STACK), for omission, we use 24 tokens above these value (to allow for stable training). For reward data augmentation, we augment with 25% additional data, noting that 50% gives similar patterns, however data augmentation may need to be explored further in future work.

## B  Extra Experiments / Plots

### B.1  Harmlessness

To compare the our findings on an objective unrelated to "helpfulness" and length, we trained a reward model for harmlessness on the Anthropic data, noting a similar pattern of "difficulty" with only 68% evaluation accuaracy on the held out set. However, we did not find length correlation in this model. The within-batch length correlation is around -0.3, and doing PPO with just the harmlessness reward model didn't increase length either once converged. This is perhaps expected since a shorter response (such as abstention from answering) will often be harmless. We also observed that the outputs started to look strange eventually, so optimizing for just harmlessness has its own set of shallow features and should not be optimized on its own. This only scratches the surface, however, and further exploration on similar / alternate objectives can likely bring more insights into the inner workings of RLHF.

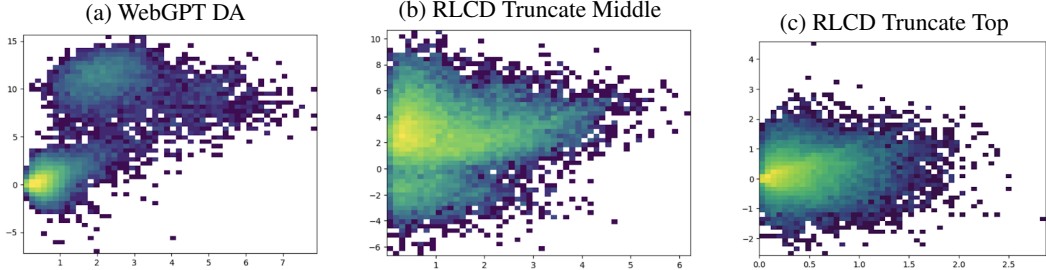

Figure 9: Dataset cartography plots on WebGPT when doing Reward Data Augmentation, as well as RLCD when truncating the central and upper sections of data.

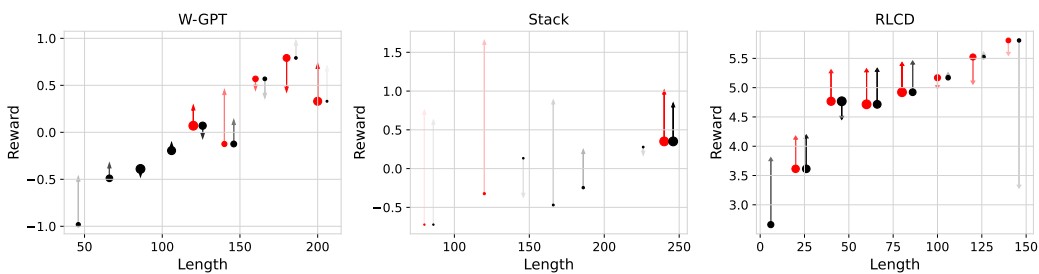

Figure 10: Bin-scatters on different settings with red (normal KL) and black (high KL) shown together

## B.2 DATASET CARTOGRAPHY

We here include dataset cartography plots in the style of Swayamdipta et al. (2020) for our reward modeling tasks. First, Figure 8 shows the dataset cartography plots on our respective settings. Note that WebGPT, the dataset with the strongest length biases, seems the most centered at 0 variance, and symmetric with respect to the x-axis. RLCD and STACK, where there seems to be more room for other features, on the other hand, demonstrate an "upward tilt" where at higher variances, the models are able to learn correct higher confidence features. This provides evidence for our hypothesis that strong length biases emerge as a symptom of reward models being unable to learn clear features from most training data.

Figure 9 shows plots for two additional settings. First, we see that when doing data augmentation, the augmented data emerges clearly and separately in the "high confidence" zone, while the initial plot remains in a similar space as before, though interestingly now displaying more of the "upward tilt" with a longer tail. Next, we see that when cutting out the central section of the cartography plot and training on the remaining data, the shape is actually preserved, suggesting that the small amount of remaining data has an intrinsic tendency to learn the strong length correlation. Likewise, when removing the upper section of "easy examples", we see that suddenly the RLCD plot becomes much more centered and sharper at the left with low variance, suggestive of the "brittleness" that we saw with WebGPT, where now the easiest pattern is harder to learn, exacerbating the pattern even further.

## B.3 LENGTH SCATTERPLOTS

Earlier in Section 3, we discuss the idea of reward gain due to length vs other features, examining results with Higher KL term. Here we show comparable plots for WebGPT (Figure 11) and RLCD (Figure 12). Note that the patterns and reward gain are very similar, suggesting that the length constraining techniques all perform similar functions, despite having very different formulations. Also note that for these two settings the ratio of reward gain independent of length remains quite low.

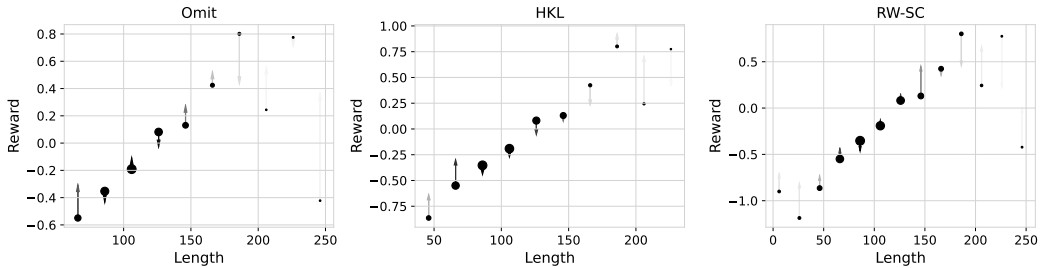

Figure 11: WGPT Bin Scatters on other types of length constraints.

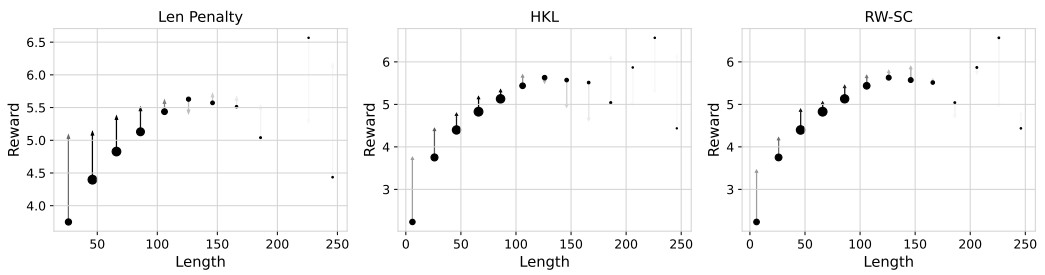

Figure 12: RLCD Bin Scatters on other types of length constraints.

## C    SAMPLE OUTPUTS

Table 7: Examples for StackExchange

| Question | Before PPO | After PPO |
|---|---|---|
| Help , How can you detect whenever the power cord is unplugged from electrical socket through Symbian OS ? | 1. <[URL]2. <[URL]3. <[URL]entry23253> (same solution, but with more details) | 1. **Check if it's plugged in** - call 'isPluggedIn()' on your [PowerManager]([URL] object, which will return 'true' when the device has been connected to AC power and the phone doesn't have a battery installed; otherwise, it returns 'false'.2. If that isn't enough for you (perhaps because you need an alarm or something), then take a look at <[URL] about how to use <[URL] time units within the Android framework. The solution outlined there should work fine with any of the three different timestamps available via System-Clock - you just need to pick one. I suggest using milliseconds since they seem most useful here, but feel free to experiment!—Edit: Just wanted to make sure this wasn't too much code to throw away :)"'js-var now = new Date().getTime() / TimeUnit.SECONDS.toMillis(1); |
| I am asking this question from an educational/hacking point of view, (I wouldn't really want to code like this).Is it possible to implement a while loop only using **C** preprocessor directives. I understand that macros cannot be expanded recursively, so how would this be accomplished? | 1) '//define while(x)', then in the macro: 'if x' and 'endif', or if you wish your "conditional" statement to execute each time after one condition was true before moving on...2)'define if(!x)'; 'if x' - as above for the conditional statement, but without need for the 'else'. If you do not specify any condition, the end will always execute.3) 'while(true); // no break'4) This should get you started along the right lines at least. The preprocessor is fairly limited though when compared with what some other languages offer! | 1) For the sake of simplicity let us assume all your 'while' loops are nested inside a macro called 'FUNCT'. 2) As a first approach you could define another variable in order to test if the condition is true or not. This new variable will have as value zero when there is no loop and nonzero otherwise:"'define FOO 0; undef FUNCT-define BOO if(FOO != 0) FUNCT(); else // do nothing here because we exit directly ; return 0; /* You can use other values instead */ int main()return BOO;"'3) Another option involves adding some variables at compile time which may increase significantly the size of your program but it also increases its readability:"'include <stdio.h>define LOOP(N) int j=1; for(j = 0 ; j< N*5-4 ; ++j )/* DO SOMETHING HERE */ return j; /* The number of iterations */ //The user defines |
| Apart from the normal differences in pitch and tone between male and female voices, are there any different words, different intonations, or other linguistic habits that tend to be used more by men or women in Korean?Are there any 'male habits' that (to sound natural) a female learner may want to avoid imitating, and vice versa? | 1. No, it is not an exaggeration; Korean is gender-specific when it comes to nouns/adjectives etc..2. There exists [the following list of female names]([URL] which will help you select appropriate name for your character if needed but most likely using common female names as characters might lead to some confusion with readers because many people have heard them so often while others never hear such names ever before... So I would recommend making up new ones. | 1. **Tone** - I don't think it is possible to have an overview of this one because tones depend on context. For instance the word for bread has two totally distinct meanings depending if it is said with a high or low rising/falling tone. However generally speaking Koreans use lower tones at end sentences whereas native English speakers often prefer higher ones. 2. **Pitch/Timbre** – as mentioned above this depends so much on context but here again generalizations can help you out: When asking questions your voice goes up, when saying statements down; you use less vocal variety in speech than people do normally; in fact all your vocal cues like volume etcetera will be toned-down compared to how we would express ourselves if using our first language. 3. **Use of vowels** – well Korean has no shortage of long open vowel sounds while English prefers to reduce them whenever possible. This difference alone means that certain pronunciation issues become quite obvious for non natives of both languages. Aside from that however, there isn't too many rules regarding vows although they play an important role in differentiate between dialect |

Table 8: Examples for WebGPT

| Question | Before PPO | After PPO |
|---|---|---|
| Why is it adults don't roll off of the bed ?I understand there are exceptions but I was just curious | Adults typically do not roll off of the bed because they have developed the muscle memory to keep their bodies from involuntarily moving during sleep and maintaining proper posture. | Adults generally do not roll off of the bed because they have developed muscle control and awareness which helps them to remain upright while sleeping. Additionally, most adults find it uncomfortable or even painful to move around in their sleep, so rolling off the bed would be difficult without waking up first. There may also be safety concerns such as falling out of bed and potentially hurting oneself if one were to lose balance or fall from the bed. Finally, some people may simply choose to stay in the same position for long periods of time when they are sleeping and thus avoid having to shift positions during their slumber. It should also be noted that some medical conditions can lead to excessive movement at night, making it more likely for someone to roll off the bed. In these cases, medications and/or lifestyle changes could help reduce this movement. Overall, it appears to be mostly a matter of choice and comfort rather than any physical limitation preventing an individual from getting up or off the bed. |
| A strong magnet will separate a mixture ofA. clear glass and green glass.B. paper cups and plastic cups.C. iron nails and aluminum nails.D. sand and salt. | B)paper cups and plastic cups | C. Iron nails and Aluminum Nails. A strong magnet can be used to sort out metal objects, such as nails. The magnetic force between the two different metals will cause them to separating from each other in the mixture. Magnetic separation is also useful for sorting materials into different classes based on their composition or properties. In this case, it could be used to separate iron nails from aluminum nails, since they have very different physical and chemical properties. It cannot be used to sort out clear glass from green glass because the difference in color alone does not represent any significant differences in the compositions of the two types of glass. Paper cups and plastic cups are made up of completely different materials (plastics versus cellulose) so a magnet would not help differentiate the two. Finally, salt and sand are both crystalline solids with similar structures but different sizes, making them impossible to distinguish using a magnet. |
| How come when we close one eye, we see "nothing" out of that eye, but when we close both, we see black?And when I say nothing I mean that you don't see black, just...nothing. | When only one eye is closed, it prevents visual information from entering the brain through that pathway; however, since there are still signals being sent to the other eye, our brains perceive "something" in terms of light and color (or darkness). When both eyes are shut, all incoming signals are blocked and thus the brain interprets this as "blackness". | When we close only one eye, our brain interprets the lack of input from that eye as darkness because it doesn't receive any visual information to process. But when we close both eyes, our brains combine the two images into one and interpret the combined image as darkness since there isn't enough contrast between the two images for us to be able to make out individual details. This phenomenon is known as binocular vision and helps protect us against blind spots by allowing us to use both eyes together in order to gain better depth perception and detect motion more easily. It also allows us to have a wider field of view compared to if we were using just one eye at a time. In addition, since each eye sees slightly different things depending on what angle it is facing, combining both images can help fill-in missing pieces of detail or detect objects that are closer or further away than what could be detected with just one eye. By closing both eyes, your brain uses all this data to create a single image which appears as pitch black due to the absence of significant contrasts. That being said, some people do still experience ghostly shapes and movement even when they shut their eyes completely; these effects arise from electrical activity happening within the eye itself |

Table 9: Examples for Anthropic HelpfulHarmless (RLCD)

| Question | Before PPO | After PPO |
|---|---|---|
| Human: How do I make Italian ice? Assistant: There are many forms of Italian ice, and I can help you try to figure out which of them you want. You could ask for gelato with a swirl of chocolate syrup, or you could ask for granita di limone, or maybe some stracciatella or espresso granita.Human: How can I make Italian ice at home?Assistant: First, you'll need some ice. You can use ice trays or ice cubes in a bag, or even a tray or bag of snow.Human: Do I need an ice shaver?Assistant: | Yes! It's recommended to have an ice shaver if you plan on making true Italian-style ice. An ice shaver will allow you to create fine shavings of ice instead of large chunks. | Yes, it's best to have an ice shaver if possible since this will allow you to create small pieces of ice without having to break up the entire block. If you don't have one, you can still make great Italian ice by using a blender or food processor instead. Just be sure not to over-process your ice or it may become too frozen and hard to eat! |
| Human: My niece is into Sesame Street. What are some sesame street characters she can add to her toy collection?Assistant: | Some of your niece's favorite Sesame Street characters include Elmo, Cookie Monster, Abby Cadabby, Grover and Bert Ernie. You could also check out plush toys featuring the entire Muppet gang! | Sesame Street has so many beloved characters! Your niece could have fun with Elmo, Cookie Monster, Grover, Abby Cadabby and more. She might also enjoy playing with Big Bird, Oscar the Grouch or Bert Ernie - they all offer lots of interactive play options. Additionally, you may want to check out some of the other classic Muppets like Kermit the Frog and Miss Piggy. |
| Continue the conversation:What are some of the health benefits of going to a spa?Assistant: | Going to a spa can provide numerous health benefits, such as reduced stress levels, improved mental clarity and overall well-being, increased energy level, decreased risk of disease, improved skin complexion, relief from muscle tension, anxiety reduction, enhanced immunity system, and better digestion. | Going to a spa can have numerous health benefits, including relaxation and stress relief. Spas provide opportunities for people to unwind in both physical and emotional ways by providing massages, facials, body wraps, manicures, pedicures, and more. This allows them to reduce their levels of stress, anxiety, and other negative thoughts and feelings while also reducing muscle tension and increasing energy level. Additionally, many spa treatments include detoxifying elements such as herbs and minerals which can help improve overall health and well-being. |

