# OpenReview forum: "A Long Way To Go: Investigating Length Correlations in RLHF"
_ICLR.cc/2024/Conference — Submitted to ICLR 2024_

### Official Review · Reviewer_McHv · 2023-10-15

**Soundness:** 4 excellent
**Presentation:** 2 fair
**Contribution:** 3 good
**Rating:** 6
**Confidence:** 4

**Summary:**

The authors show that when applying RLHF for optimize helpfulness, much if not most of the gain comes simply from learning to increase the length of the response, and in fact solely optimizing for length can achieve most of the gains of optimizing for helpfulness in many cases.

**Strengths:**

--comprehensive evaluations on a few different RLHF settings/datasets for helpfulness optimization

--nice in-depth exploration confirming the extent of a phenomenon (length correlations for helpfulness optimization) that i think hasn't been comprehensively analyzed previously

**Weaknesses:**

--presentation: it's quite difficult to parse what several of the abbreviations mean in the tables (and honestly i'm still a bit confused on what exactly you're showing in the first couple of figures and tables after looking at them for several minutes). it would be great if you could make the figs/tables more self-contained, e.g., define abbreviations in the captions and summarize takeaways, so i don't have to keep bouncing between the figs/tables and the text.

**Questions:**

--even if we're just optimizing for length, your model still learns something nontrivial, right? an interesting point of comparison might be an extra baseline where instead of sampling 8 outputs from the base LM and picking the longest one, you just sample one output and forcibly make it keep generating past the end - i imagine this would perform a lot worse. so intuitively, i feel that optimizing for length is also somehow optimizing for some less trivial notion of comprehensiveness, maybe?

--just to confirm, weighted reward gain i should think of as measuring how much reward improves within a particular bin of length, i.e. the reward gain that comes from factors other than length?

--would be curious also if you have any qualitative sense of what are the non-length axes along with helpfulness improves in your experiments (i.e., any intuitions for the remaining 10-30% of helpfulness that wasn't attributed to increased length?).

--as you wrote in the conclusion, it would be interesting to just see a quick experiment on e.g., harmlessness or honesty, if only to confirm our intuition that length is primarily a heuristic for optimizing helpfulness and not really applicable to other common tasks we RLHF for. (not necessary if it's troublesome to setup though.)

--i'm not very clear on the concrete actionable suggestions for practitioners based on your analysis? (though they're not strictly necessary - i see that you did try a few ways to regularize length already)

---

> ### Author Response · Authors · 2023-11-18
> **Response to Reviewer McHv**
>
> Thanks for the feedback and interesting questions!
>
> > presentation: it's quite difficult to parse what several of the abbreviations mean in the tables
>
> Thanks so much for this feedback, we uploaded a new version with figures and tables that we simplify and tie more clearly to presentation in the text, as well as clarifying experiments further to be easier to read.
>
> > concrete actionable suggestions for practitioners based on your analysis? (though they're not strictly necessary - i see that you did try a few ways to regularize length already)
>
> Thanks for asking! Some concrete suggestions are: (1) As you suggested, if one simply wants to get most of the “improvement” of RLHF without dramatic length increases, several of our proposed interventions are actually good options. (2) Downstream evaluation isn’t a gold standard for RLHF learning good features. Currently, better human/simulated preferences are often assumed to indicate that RLHF is learning “desired” features, dismissing further investigation. Especially if it’s high-risk (medical, legal application, etc.) RLHF practitioners should measure success in highly-specific terms, to validate assumptions about features learned. (3) RLHF work should focus mostly on reward modeling (better objective functions, collecting better data, engineering to enable larger scale with less compute, etc.). This direction is relatively underexplored, and will likely be the first step to substantial improvements
>
> > even if we're just optimizing for length, your model still learns something nontrivial, right?
>
> We thought that this was really interesting too, our hypothesis is that the joint optimization of length with the KL penalty is what leads to the “non-trivial” learning of features. Since repetitive, pathological outputs, etc. would likely have a higher KL divergence from the initial policy, this term likely forces the model to learn how to generate more descriptive outputs while ALSO maximizing length. This is probably why the lambda=0 baseline is sometimes much worse than the standard lambda=0.04 counterpart for pure length optimization.
>
> >  it would be interesting to just see a quick experiment on e.g., harmlessness or honesty, if only to confirm our intuition that length is primarily a heuristic for optimizing helpfulness and not really applicable to other common tasks we RLHF for. (not necessary if it's troublesome to setup though.)
>
> We trained a reward model for harmlessness on the Anthropic data. We did not find length correlation in this model. The within-batch length correlation is around -0.3, and doing PPO with just the harmlessness reward model didn’t increase length either once converged. This is perhaps expected since a shorter response (such as abstention from answering) will often be harmless. We also observed that the outputs started to look strange eventually, so optimizing for just harmlessness has its own set of shallow features and should not be optimized on its own. Thanks for the suggestion!
>
> > just to confirm, weighted reward gain i should think of as measuring how much reward improves within a particular bin of length
>
> Yes, weighted reward gain gives an aggregate notion of reward improvement purely within bins
>
> > Non-length axes for remaining 20-30%
>
> Thanks for asking! We thought a bit about this too. Based on looking at outputs and RM scores from before and after RLHF on some different settings, some alternate axes may be related to things like answer relatedness to the question (completely unrelated outputs will get lower scores, off-topic generation seems to go down from RLHF), and grammaticality. That said, a more comprehensive study of qualitative changes from RLHF (especially once we have more robust reward models in the future), would be interesting for future work.

---

> > ### Comment · Reviewer_McHv · 2023-11-22
> > **Thanks for your response**
> >
> > Thanks for answering my questions. I'd encourage you to also update your paper with some of these discussions, and add your new harmlessness experiment to the Appendix too (I don't think I saw it in there?).

---

> > > ### Author Response · Authors · 2023-11-23
> > > **Follow-up response to Reviewer McHv**
> > >
> > > Thanks for responding and the additional feedback! We went ahead and added the harmlessness experiment to the appendix. When we have more space in a future version, we can definitely also make sure to incorporate more discussion, particularly on the non-length features learned both by RMs and PPO, and on concrete suggestions.

---

### Official Review · Reviewer_rbH1 · 2023-10-28

**Soundness:** 3 good
**Presentation:** 2 fair
**Contribution:** 3 good
**Rating:** 6
**Confidence:** 4

**Summary:**

This paper explores how length correlates with performance improvement in RLHF. The author starts by examining the output length increase in RLHF and the correlation between output length and reward model scores, and then they reveal that some intervention in PPO can mitigate the length increase to some extent. Further, they find that the reward model can only learn "easy" examples, i.e. the examples that can be correctly predicted according to the length heuristic. The authors examine several intervention methods for reward modeling and find that there could be a trade-off between length bias and reward model accuracy in some cases. Finally, they show that purely optimizing the policy model for increasing output length can lead to an improvement in output quality.

**Strengths:**

This paper focuses on a very important problem: what role does length play in RLHF? The paper conducts extensive experiments to demonstrate the correlation between length and reward model scores and explores several ways to mitigate length bias. The results can provide constructive guidance for future research.

**Weaknesses:**

1. The most confusing part for me is the evaluation. What is the rationale for adopting a length-bias metric, GPT-4 evaluation [1], when evaluating the correlation between length and RLHF performance? Two potential factors can be affected by length, making it hard to disentangle the attribution of length correlation. Or do you aim to reveal the bias issue of GPT-4 in this paper? Then could you please explain more clearly what you mean by "length correlations in RLHF" and what the length correlates with? The reward modeling? The optimization algorithm? The evaluation? Or all of them? If so, I'd suggest investigating them separately.
2. I am skeptical about the experiment and claim that "reward models are not able to make progress on most training examples" (sec 4.1). The results may, at least partly, be due to the reward model capacity. I suppose that reward modeling is a relatively difficult task, so weak models may only capture some shallow features. With a more capable reward model, the "hard" samples may be learned. A supporting evidence is that, as shown in Table 3, the length heuristics accuracy on WebGPT is 55.7%. However, in the result of a recent paper [2], a more powerful reward model that is not trained on WebGPT can yield 65.2% accuracy. Therefore, according to this paper, at least 9.5% of counter-biased samples are learned by the reward model. Perhaps the authors can supplement experiments by adopting more powerful reward models, e.g. scaling the base from LLaMA-7B to 70B following the original LLaMA-2 paper [3] (if computation allowed), and see if the learning patterns remain the same.

References:
[1] AlpacaFarm: A Simulation Framework for Methods that Learn from Human Feedback. Dubois et al.

[2] UltraFeedback: Boosting Language Models with High-quality Feedback. Cui et al.

[3] Llama 2: Open Foundation and Fine-Tuned Chat Models. Touvron et al.

**Questions:**

Please refer to weaknesses.

---

> ### Author Response · Authors · 2023-11-18
> **Response to Reviewer rbH1**
>
> Thanks for the constructive feedback and questions!
>
> > Could you please explain more clearly what you mean by "length correlations in RLHF" and what the length correlates with? The reward modeling? The optimization algorithm? The evaluation? Or all of them? If so, I'd suggest investigating them separately.
>
> We examine “length correlations” in this paper in the following contexts:
> - Preference data: Is label distribution imbalanced to favor length?
> - Reward modeling (Section 4): Does reward score correlate with length? Does “easiness” of examples correlation with length? (connects to preference data)
> - PPO (Section 3.5): Does PPO lead to longer outputs if we do things like increase the KL penalty?
> Since these are different notions, we generally dedicate different sections / experiments to each of these components, but we can make sure to clarify these distinctions further. Importantly, we *don’t* seek to analyze length bias within evaluation, since our hypotheses largely don’t depend on GPT-4 evaluation.
>
> > The most confusing part for me is the evaluation. What is the rationale for adopting a length-bias metric, GPT-4 evaluation [1], when evaluating the correlation between length and RLHF performance? Two potential factors can be affected by length, making it hard to disentangle the attribution of length correlation. Or do you aim to reveal the bias issue of GPT-4 in this paper?
>
> Yes, both GPT-4 and human evaluators likely have a bias towards longer responses (recall our goal isn’t to reveal these evaluation biases), but we find they can still reveal useful information when length biases are controlled (see claims below). For comparability with prior work, we follow the protocol of [3] for “simulated preference” downstream evaluation, as it was shown to correspond well with human preference.
>
> We use simulated preferences as follows:
> - Context: RLHF generally makes “numbers go up” relative to SFT on standard settings (Table 2), which largely follows prior work
> - Claim: Reward interventions can improve quality of outputs independent of length (Table 5)
> - Claim: Length-only PPO is both better than a baseline of sampling longer outputs, and comparable to standard PPO (Table 6).
>
> Importantly, notice that for both claims, the simulated win rates are actually higher for models with much shorter lengths on average: these results are not explained away by saying that GPT-4 prefers longer outputs in simulated win rate experiments.
>
> Finally, note that most of our results (Fig 2,3,4; Table 1,2,3,4) are based on intrinsic measures like reward model scores and output length, which are unaffected by any biases from GPT-4.
>
> > I am skeptical about the experiment and claim that "reward models are not able to make progress on most training examples" (sec 4.1). The results may, at least partly, be due to the reward model capacity. [...] Perhaps the authors can supplement experiments by adopting more powerful reward models [...]
>
> Thanks for mentioning this! We actually agree with this point. We didn’t intend to claim that reward models *in general* couldn’t learn from most examples on those datasets, but rather that this happens with the 7B scale reward models with current objectives and data, then propagating length correlations throughout RLHF. This is why a big takeaway for future work is to focus on improving reward modeling especially at more scientifically feasible scales.
>
> We did our experiments with 7B-sized models largely due to hardware limitations (because PPO involves storing several large models). Note that:
> - The majority of open RLHF work uses 7B, or often even smaller-scale models [1][2][3][4].
> - *All* our trained 7B reward models actually outperform length heuristic (61.5% on WebGPT vs 55.7), and thus are likely learning other features, yet PPO still largely optimizes for length.
> - Regarding UltraFeedback (the paper with 65.2% accuracy on WebGPT, which we note was arxived since the ICLR deadline), scale may be a component, but they also use very different (potentially better) preference data. This supports our overall message of needing to improve preference data and reward modeling
>
> That said, based on your feedback, we scaled our RM experiments up to LLaMA-2 13B. We report intrinsic reward modeling accuracy results below (compare with Table 4 in the paper):
>
> | Model | WebGPT | Stack | RLCD |
> |-------|--------|-------|------|
> | LLaMA 7B    | 61.5%  | 70%   | 80%  |
> | LLaMA-2 13B   | 64.5%  | 71.3% | 81.2%|
>
> As shown in the table, we found that reward modeling accuracy is only marginally better than it was before, suggesting that model scale likely isn’t the main bottleneck on the “hard” examples. We agree it would be good to scale up these results further, but we think the main takeaway of our results and those in UltraFeedback is the need for better datasets rather than just scaling.
>
> Thanks again for the experiment suggestion!

---

> > ### Comment · Reviewer_rbH1 · 2023-11-20
> > **Response to Authors**
> >
> > Thanks for the clarification. I would like to raise my scores, but I hope that the authors can improve the presentation of this paper in the revision.

---

> > > ### Author Response · Authors · 2023-11-20
> > > **Follow-Up Response to Reviewer rbH1**
> > >
> > > Thanks for responding! Just to double check, did you get the chance to look at the updated pdf we uploaded? We made some changes which we believe improve the clarity.

---

> ### Author Response · Authors · 2023-11-18
> **Response Citations**
>
> [1] Simeng Sun, Dhawal Gupta, and Mohit Iyyer. Exploring the impact of low-rank adaptation on the performance, efficiency, and regularization of RLHF. ArXiv, abs/2309.09055, 2023.
>
> [2] Kevin Yang, Dan Klein, Asli Celikyilmaz, Nanyun Peng, and Yuandong Tian. RLCD: Reinforcement Learning from Contrast Distillation for Language Model Alignment.
>
> [3] Yann Dubois, Xuechen Li, Rohan Taori, Tianyi Zhang, Ishaan Gulrajani, Jimmy Ba, Carlos Guestrin, Percy Liang, and Tatsunori B. Hashimoto. AlpacaFarm: A Simulation Framework for Methods that Learn from Human Feedback, 2023.
>
> [4] Zeqiu Wu, Yushi Hu, Weijia Shi, Nouha Dziri, Alane Suhr, Prithviraj Ammanabrolu, Noah A. Smith, Mari Ostendorf, and Hanna Hajishirzi. Fine-grained human feedback gives better rewards for language model training. ArXiv, abs/2306.01693, 2023.

---

### Official Review · Reviewer_tqJD · 2023-10-30

**Soundness:** 3 good
**Presentation:** 2 fair
**Contribution:** 2 fair
**Rating:** 6
**Confidence:** 3

**Summary:**

This paper studies whether RLHF scores are correlated with the length of responses. Given that length preference is a well-known issue in the RLHF literature, in the work, the authors explore how much of the optimization and improvement in RLHF is based on length, as opposed to other factors.

Using a comprehensive set of experiments, the paper shows that length constitutes the majority of the reward, indicating that length may play a much larger role than previously documented.

Finally, the authors discuss the implication of these findings on the RLHF and state the importance of developing better and more robust algorithms.

**Strengths:**

The paper offers a well-executed investigation of a well-known pattern: the correlation between RLHF scores and length. The findings are persuasive and supportive of the conclusions

**Weaknesses:**

The paper's comprehensibility is somewhat challenging. The conclusions drawn from the tables and figures lack clarity, and it's not easy to discern the key takeaway from the experiments presented. The paper would be improved with some rewriting and clarification.

The experiments themselves are well-executed, and they do support the main message. However, it's worth noting that this pattern has been observed in numerous other studies, and strategies to address this bias/reward hacking have been extensively documented elsewhere. Consequently, the contribution of this paper does not look very strong for an ICLR  conference.

**Questions:**

NIT
* The "STD" term was introduced in Figure 2 way before defining it later in the text.
* For Figure 3, why not have both settings (red and black) for all the lengths? Does that mean that sometimes there are no examples in a particular bin for a particular setting or are there some missing datasets in the figure?

---

> ### Author Response · Authors · 2023-11-18
> **Response to Reviewer tqJD**
>
> Thanks for the thoughtful comments!
>
> > a well-known pattern: the correlation between RLHF scores and length
> > this pattern has been observed in numerous other studies, and strategies to address this bias/reward hacking have been extensively documented elsewhere. Consequently, the contribution of this paper does not look very strong for an ICLR conference.
>
> We provide the most in-depth study of this phenomenon to date, indicating that length may play a much larger role than previously anticipated by the community. Length increases from RLHF have been observed previously (e.g., in “Secrets of PPO”, which we cite), but we believe our analysis, and the new tools we introduce, contribute a lot to our understanding beyond just observing the pattern.
>
> For instance, we conclude the following, which are not documented in prior work:
> - PPO’s reward optimization strategy relies *substantially* on length at the cost of other features
> - PPO’s overall reward optimization strategy is qualitatively similar across various RL interventions given the same reward model: RMs are the most crucial component
> - Reward robustness issues are the primary source of length biases; current open reward models struggle on preference data, and *heavily* rely on “easy” subsets of training data with shallower features
> - Doing PPO with *just length* is comparable to full RLHF with trained reward models
>
> If there are additional papers we should cite or contextualize our work with respect to, we’d love any pointers!
>
> > Feedback on clarity, STD term
>
> Thanks for pointing these out! We made a further pass through the paper to clarify things, especially key take-aways from experiments, and improve captions / Figure 3 based on your feedback.
>
> > For Figure 3, why not have both settings (red and black) for all the lengths? Does that mean that sometimes there are no examples in a particular bin for a particular setting
>
> Yes. Due to length shifts, there are cases where, for example, PPO doesn’t generate any outputs at shorter lengths, leading to us being unable to have a comparative bin (which is one of the reasons why we use a length-constrained high-KL version to better validate our hypotheses in a case with more overlapping bins).
>
> We realized that putting both versions in the plot was a bit overloaded, so we made a simpler version with only the high-KL arrows, keeping both normal and high-KL numbers in the WRG table and moving the 2-color plot to the appendix. Thanks for the feedback!

---

### Official Review · Reviewer_PX4u · 2023-11-01

**Soundness:** 4 excellent
**Presentation:** 3 good
**Contribution:** 3 good
**Rating:** 6
**Confidence:** 3

**Summary:**

The paper investigates the length corrections with the RLHF paradigm for language model (LM) training. Specifically, it has been observed that models finetuned with RLHF tend to generate longer sentences. In this paper, the authors study the causes of the phenomenon through multiple aspects, including the RL algorithm (i.e., PPO) for training the LM, the reward models (RM), and preference data. They demonstrate the PPO generally increases the length compared to the supervised model regardless of the reward engineering applied to mitigate the issue. In addition, altering the data for reward model training does not fully solve the issue either. Lastly, they show that solely optimizing for the length recovers most of the performance improvements.

The highlight of the paper is it clearly documents the length-increasing issue in the RLHF pipeline. They conduct experiments on several datasets across different domains to demonstrate the issue.

However, the paper is rather descriptive than prescriptive. Specifically, the authors describe the correlation between length increasing and standard PPO training without providing the underlying reason for the phenomenon. Although they propose several heuristic-inspired remedies, the problem is not fully resolved. Therefore, it might not directly contribute to improving the existing RLHF method.

**Strengths:**

The paper clearly documents the length-increasing issue in the RLHF pipeline. They conduct experiments on several datasets across different domains to demonstrate the issue.

**Weaknesses:**

The paper is rather descriptive than prescriptive. Specifically, the authors describe the correlation between length increasing and standard PPO training without providing the underlying reason for the phenomenon. Although they propose several heuristic-inspired remedies, the problem is not fully resolved. Therefore, it might not directly contribute to improving the existing RLHF method.

**Questions:**

N/A

---

> ### Author Response · Authors · 2023-11-18
> **Response to Reviewer PX4u**
>
> Thanks for the thoughtful comments! We’ll try to address concerns below:
>
> >The paper is rather descriptive than prescriptive… Although they propose several heuristic-inspired remedies, the problem is not fully resolved. Therefore, it might not directly contribute to improving the existing RLHF method.
>
> We agree that the problem is not fully resolved. Length correlation is difficult to fully resolve since longer responses *are* in some cases more informative! So we do not believe it is possible to provide a crisp solution here without building new preference datasets. Instead, our main goal is to characterize the phenomenon and possible solutions. We explore “heuristic-inspired remedies” precisely because they are simple solutions that practitioners will reach for, and shed light on how different components of RLHF influence the observed patterns.
>
> > Specifically, the authors describe the correlation between length increasing and standard PPO training without providing the underlying reason for the phenomenon.
>
> We believe that we do provide reasons. The initial length stratification and intervention-based experiments focus on studying how PPO optimizes reward, demonstrating that reward model biases are the main cause behind length increases in PPO. We then dedicate Section 4 to analyzing in depth the underlying reasons for those biases; we discover the main culprits to be imbalances in preference data, brittleness in the reward modeling objective, and “easy” examples from our dataset cartography analysis.

---

### Author Response · Authors · 2023-11-18
**General Response to Reviewers**

Thanks to all reviewers for the thoughtful and constructive feedback!

Beyond our individual responses below, we have uploaded a new version of the paper which we believe addresses some of the concerns:
- Updated abstract and intro to better contextualize the takeaways of experiments
- New Figure 3 with less overloaded content (focusing on the high-KL variant with more overlapping bins) to make the stratification experiment easier to understand.
- New captions in results figures / tables explicitly stating key takeaways from each experiment
- Better linking between acronyms in figures, captions, text

We also ran further experiments to answer reviewer questions:
- 13B reward model experiments (see response to Reviewer #rbH1)
- Harmlessness setting (see response to Reviewer #McHv)

---

### Meta-Review · Area_Chair_buDS · 2023-12-06

**Metareview:**

The paper investigates the correlation between the length of responses and the performance of Reinforcement Learning from Human Feedback (RLHF) in language model training. The authors conduct experiments on several datasets and find that length constitutes the majority of the reward, indicating that length may play a much larger role than previously documented.

The reviewers have acknowledged the comprehensive nature of the experiments and the clear documentation of the issue of length increase in the RLHF pipeline. However, they have also pointed out that the paper tends to be more descriptive than prescriptive, and there is a need for more clarity in the presentation of results, and not contributing significantly to the existing body of knowledge on RLHF. The major concern for me is the conclusion and contribution of this work is vague. The dominance of length in RLHF is related to too many aspects: human's natural preference of long texts, the quality of short/long texts in training data, etc. The paper proposed an interesting observation but did not provide a successful explanation to the root of this observation.

I think this work is potentially a valuable contribution to the field, but the current version of the paper needs significant improvements.

**Justification For Why Not Higher Score:**

While the paper presents an interesting investigation into the correlation between response length and RLHF performance, it falls short in several key areas.

The paper is more descriptive than prescriptive, identifying the issue but not providing a comprehensive solution or a clear understanding of the underlying reasons for the phenomenon.

The presentation of results is also somewhat unclear, making it difficult to discern the key takeaways from the experiments.

Furthermore, the paper's contribution to the existing body of knowledge on RLHF is not limited.

**Justification For Why Not Lower Score:**

N/A

---

### Decision · Program_Chairs · 2024-01-16

Reject